# DenseTrack: Drone-based Crowd Tracking via Density-aware Motion-appearance Synergy

## ABSTRACT

Drone-based crowd tracking faces difficulties in accurately identifying and monitoring objects from an aerial perspective, largely due to their small size and close proximity to each other, which complicates both localization and tracking. To address these challenges, we present the Density-aware Tracking (DenseTrack) framework. DenseTrack capitalizes on crowd counting to precisely determine object locations, blending visual and motion cues to improve the tracking of small-scale objects. It specifically addresses the problem of cross-frame motion to enhance tracking accuracy and dependability. DenseTrack employs crowd density estimates as anchors for exact object localization within video frames. These estimates are merged with motion and position information from the tracking network, with motion offsets serving as key tracking cues. Moreover, DenseTrack enhances the ability to distinguish small-scale objects using insights from the visual-language model, integrating appearance with motion cues. The framework utilizes the Hungarian algorithm to ensure the accurate matching of individuals across frames. Demonstrated on DroneCrowd dataset, our approach exhibits superior performance, confirming its effectiveness in scenarios captured by drones.

## CCS CONCEPTS

• **Computing methodologies** → **Tracking**; Computer vision; • **Human-centered computing** → *Collaborative and social computing*.

## KEYWORDS

Multiple object tracking, Crowd Localization, Vision-language pre-training, Motion-appearance Fusion

**ACM Reference Format:**

Anonymous Author(s). 2024. DenseTrack: Drone-based Crowd Tracking via Density-aware Motion-appearance Synergy. In *Proceedings of the 32nd ACM International Conference on Multimedia (MM'24), October 28-November 1, 2024, Melbourne, Australia.* ACM, New York, NY, USA, 9 pages. https://doi.org/10.1145/nnnnnnn.nnnnnnn

## 1 INTRODUCTION

Drone-based crowd tracking leverages unmanned aerial vehicle (UAV) cameras for automated surveillance, playing a critical role in crowd management and monitoring. This technology is designed to

**Unpublished working draft. Not for distribution.**

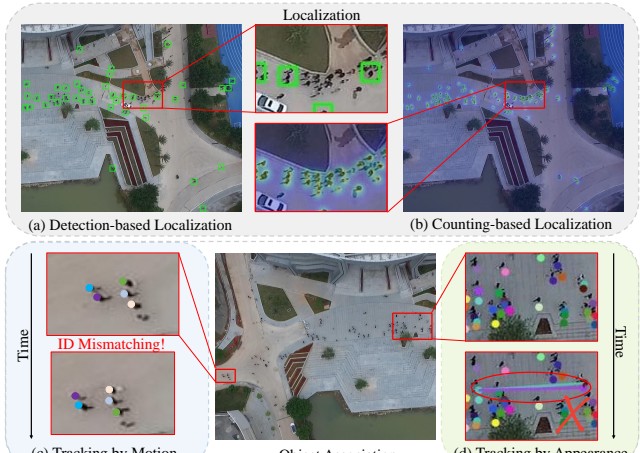

**Figure 1: Illustration of localization and tracking techniques. The upper section contrasts (a) detection-based localization, which relies on identifying objects directly, with (b) counting-based localization, which estimates object positions through density analysis. The lower section highlights inaccuracies in (c) Tracking by Motion, where predictions are based on movement patterns, and (d) Tracking by Appearance, which uses visual features; identically colored points indicate predictions for the same individual.**

identify and consistently track individuals across successive video frames, amidst ongoing movement of both the subjects and the background. To achieve this, the process of multi-object tracking (MOT) is utilized [5, 7, 11, 24, 31], which involves two critical steps: localization and tracking. Localization discerns the exact positions of objects within each frame, while tracking maintains consistent identification of these objects over time. These tasks are complicated by factors such as object size, density, and environmental complexity. Fig. 1 depicts the various approaches to localization and tracking, showing the challenges each method faces.

Regarding the localization task, Figs. 1(a) and (b) depict the performance of detection-based and counting-based methods. Detection methods struggle with small objects and complex backgrounds, often resulting in significant errors. Conversely, counting-based methods provide almost accurate localization of all target individuals in densely populated scenes. However, unlike detection methods, counting approaches sacrifice a considerable amount of individual appearance information, which complicates the use of similarity-based tracking techniques in MOT with bounding boxes. This loss presents a substantial challenge in balancing precise localization with the preservation of appearance information for individuals.

For the tracking task, as shown in Figs. 1(c) and (d), the methods of tracking by motion and tracking by appearance are explored. Tracking by motion effectively considers the inter-frame movement of objects but can lead to misidentification in scenarios with dense,

small targets. On the other hand, tracking by appearance, while focusing solely on the visual attributes, often mistakenly associates distant and different objects due to ignoring inter-frame motion. Thus, the second challenge involves effectively using inter-frame physical distances to minimize these errors.

In addressing the first challenge, various counting-based tracking methods have been developed to balance precise localization with the preservation of appearance information for individuals from drone perspectives. For instance, STNNet [41] leveraged density maps for crowd localization and motion offsets for tracking. Although this method significantly enhances localization accuracy, it struggles with object displacement issues, particularly due to the small size and close proximity of objects in aerial views. Additionally, the multi-frame attention-based method [3] aims to improve tracking by integrating features across multiple frames. However, its dependency on consecutive frames reduces its effectiveness in scenarios characterized by large inter-frame intervals.

Addressing the second challenge involves MOT methods [9, 10, 25, 30] that combine appearance and motion cues to capitalize on their strengths and mitigate errors. However, tracking in drone-based environments presents unique difficulties, particularly with the detection of small objects. Extracting individual appearance features from density maps in drone perspectives is notably challenging and less effective compared to detection-based methods, which inherently capture richer detail, whereas density maps offer limited information.

In this paper, we introduce the Density-aware Tracking (*DenseTrack*) framework, which advances the counting-based localization framework by incorporating both motion and appearance cues. DenseTrack tackles two critical tasks: extracting detailed appearance information from density maps for precise individual identification and correcting motion discrepancies using this appearance data. Initially, DenseTrack utilizes visual-language models (VLMs) to derive intricate appearance features from density maps, ensuring accurate characterizations of individuals. The appearance data thus extracted is then seamlessly integrated with motion and position data to address motion inaccuracies, enhancing the fidelity of motion cues. This strategic integration effectively surmounts the challenges of object localization in drone-based scenarios, while adeptly merging both motion and appearance information into the tracking process.

In summary, our contributions to the field are threefold:

- We introduce the Density-aware Tracking (DenseTrack) framework, a novel approach that synergistically combines motion and appearance cues within a crowd counting localization paradigm. This strategy effectively exploits the strengths of both cues while mitigating their limitations.
- We enhance the process of individual identification within density maps by integrating a visual-language model. This integration significantly improves the descriptive capabilities of density maps, enabling more nuanced and accurate representations of individuals in crowded scenes.
- We demonstrate the superior performance of our approach using DroneCrowd dataset, where it outperforms existing methods in the field of crowd tracking.

## 2 RELATED WORK

**Crowd Counting** is essential for effective crowd management and has received significant attention in recent years. It can be broadly classified into three categories: detection-based methods [2], regression-based methods, and density map-based methods [8, 40, 44, 52]. While detection-based methods struggle in densely populated scenes, regression-based approaches often fail to localize individuals accurately in sparser crowds, making density map-based methods the preferred technique. This approach has proven superior to traditional methods, demonstrating exceptional efficacy. Previous solutions, such as multi-branch networks [48], aimed to address the varying scales of crowd distribution but typically produced suboptimal density maps. The introduction of null convolution has revolutionized this area by preserving pixel information and reducing parameter count, thereby enhancing performance. The evolution of deep learning has further expanded and improved the architecture of backbone networks. The strategic development of convolutional neural networks (CNNs) and the incorporation of Transformer networks into single-domain approaches have become increasingly effective [20, 26, 29, 36]. Recent innovations have even enabled precise crowd localization [12, 21, 43, 51]. Despite these advancements, solely focusing on crowd numbers is insufficient for comprehensive crowd management. Assessing crowd movement is also important for identifying potential risks within a crowd.

**Multi-object Tracking** poses a considerable challenge in computer vision, involving the detection and continuous tracking of multiple objects across video sequences [1, 33, 34, 39, 50]. Traditionally, methods such as active contours [14], particle filters [46], and various association techniques [23, 37, 47, 49] have been employed. However, there has been a significant shift towards a tracking-by-detection paradigm in recent years. This approach uses bounding-box detectors to identify objects and leverages appearance features for association, although it often struggles with accurately detecting smaller objects due to their lack of distinctive features.

The Simple Online Real-Time Tracker [6] provided an efficient solution for MOT, featuring rapid update frequencies and minimal processing requirements. Building on SORT, DeepSort [42] incorporated deep learning-based association metrics to significantly enhance tracking accuracy by using more sophisticated data association techniques. Additionally, Zhang et al. [47] developed ByteTracker, an advanced tracking algorithm that utilizes deep neural networks. ByteTracker is noted for its exceptional accuracy and robust performance in challenging environments, making it a powerful tool for complex MOT tasks.

**Crowd Tracking** has witnessed significant advancements, with innovative developments reshaping the field. Kratz and Nishino [16] utilized a space-time model to track individuals within crowds effectively. AdaPT [4] introduced a real-time algorithm that deduces individual trajectories in dense environments, enhancing the understanding of crowd dynamics. Recent methods such as tracking-by-counting [28] integrated detection, counting, and tracking to leverage complementary data, proving to be effective for real-time people counting applications [32]. Furthermore, Sundararaman et al. [35] developed the Congested Heads Dataset, which combines a

head detector with a Particle Filter and a re-identification module to efficiently track multiple individuals in crowded settings.

## 3 DENSETRACK

### 3.1 Problem Formulation

Focusing on small, densely packed objects, this paper introduces a counting-based method for drone-based crowd tracking, integrating appearance and motion cues to compensate for their respective limitations. The framework, depicted in Fig. 2, comprises three stages: Localization, Individual Representation, and Object Association and Tracking. The input consists of all frames $I = \{I_1, I_2, \cdots, I_N\}$ from the video stream $V$, where $N$ denotes the total number of frames. The output includes trajectories $T = \{T_1, T_2, \cdots, T_M\}$ for each individual within the video stream $V$, with $M$ denoting the total number of individuals detected.

The Localization stage involves sequentially inputting all frames $I$ from the video stream $V$ into the crowd counting network (CN) to derive the coordinate list $CL = \{CL_1, CL_2, \cdots, CL_N\}$ for each frame image, given by:

$$CL_i = CN(I_i), \quad (0 \le i < N).$$ (1)

In the Individual Representation (IR) stage, all frames $I$ from the video stream $V$, along with the coordinate list CL$i$ of individuals in each frame $I_i$, are inputted. Then, leveraging the localization from density maps, we obtain estimated positions $\tilde{CL}_{i-1}$ from the last frame $I_{i-1}$ and appearance representations $F_i$ of individuals in each frame $I_i$. The formula is defined as:

$$\tilde{CL}_{i-1}, \hat{F}_i = IR(I_i, CL_i).$$ (2)

In the final stage, Object Association and Tracking (OAT), individuals' appearance representations $F = \{F_1, F_2, \cdots, F_N\}$ function as appearance cues, while the estimated positions $\tilde{CL} = \{\tilde{CL}_1, \tilde{CL}_2, \cdots, \tilde{CL}_{N-1}\}$ and coordinate list CL serve as motion cues. This stage entails matching individuals across different frames, culminating in the derivation of individual trajectories $T$ as:

$$T = OAT\left(\tilde{CL}, CL, \hat{F}\right).$$ (3)

### 3.2 Localization

Localization forms the basis of tracking. Given detectors' limitations in identifying small objects from the high-altitude overhead perspective of drones, it's essential to establish a solid tracking foundation. Thus, we introduce crowd counting network localization as a replacement for traditional detection networks. Specifically, we input all frames $I$ of the video stream $V$ frame by frame to obtain their corresponding density maps $D = \{D_1, D_2, \cdots, D_N\}$.

However, the prevalent issue with widely used density maps, lacking precise individual localization, impedes accurate crowd localization. Inspired by [21], we employ the focal inverse distance transform map as the density map and use the high-resolution network (HR) for density map prediction:

$$D_i = HR(I_i), \quad (0 \le i < N).$$ (4)

After obtaining the density map $D$ for each frame of the video, each pixel in the density map signifies the likelihood of an individual's presence. Consequently, in this density-aware stage, if a point on the density map is a local maximum (LM), the coordinates of that point are considered as the coordinates of an individual in the frame. Thus, the coordinate list of all individuals in each frame is derived, denoted as $CL_i$:

$$CL_i = LM(D_i), \quad (0 \le i < N).$$ (5)

### 3.3 Individual Representation

After obtaining accurate positions of individuals in each frame, extracting effective representations for inter-frame association is crucial. To integrate both appearance and motion information, the simultaneous extraction of both appearance features $F$ and motion offsets $\vec{o}$ is adopted as association cues.

*3.3.1 Appearance Representation.* Considering the inherent limitations of density maps in providing detailed individual information and the critical role of rich appearance features in tracking accuracy, we proceed to acquire appearance representations $F$ for all individuals in the frames.

For unsupervised extraction of individual representations, we employ the vision-language pre-training model, BLIP2 [17]. By utilizing Cut, the original images are cropped based on individual localization, extracting local patches representing individuals in each frame. Denoted as $cl_{i,j} \in CL_i (0 \le j < M_i)$, where $M_i$ represents the number of individuals appearing in the $i$-th frame, these individuals are then used to obtain sub-images $S_{i,j}$ corresponding to each individual:

$$S_{i,j} = Cut\left(cl_{i,j}, I_i\right).$$ (6)

After obtaining individual local patches $S_{i,j}$ for each individual in every frame, BLIP2's feature extraction (BE) module is employed to acquire appearance representations $F_{i,j}$ for each individual:

$$F_{i,j} = BE\left(S_{i,j}\right),$$ (7)

where the representation $\hat{F}_{i,j}$ obtained here is a matrix with dimensions $(W, H)$, which is not convenient for merging all individual identifiers in subsequent frames. Therefore, we flatten the matrix $F_{i,j}$ to obtain $\hat{F}_{i,j}$, with dimensions $(1, W \times H)$.

*3.3.2 Motion Representation.* In the Motion Representation stage, accurately determining the motion offsets of individuals in dense small object scenes is crucial. We utilize density maps to locate individuals within the frame $CL_{i,j}$ and gather the corresponding motion information at these positions. However, density maps lack motion offsets $\vec{o}_{i,j}$ and are limited to counting and localization. Thus, motion information extraction is necessary as density maps lack individual motion offsets.

Inspired by [13], we utilize the motion and position map (MPM) to predict the motion states of individuals. Specifically, given frames $i$ and $i + 1$, we generate MPM $C_{i+1}$. In MPM $C_{i+1}$, the value $C_{i+1,j}$ at each pixel point where individual $j$ is located is calculated based on the motion offset of that individual as follows:

$$C_{i+1,j} = G(p_t) \frac{cl_{i,j} - cl_{i+1,j}}{\left\|cl_{i,j} - cl_{i+1,j}\right\|_2},$$ (8)

where $G(p_t)$ is specifically derived through Gaussian filtering and signifies the likelihood that the point corresponds to an individual.

Then, frames $I_i$ and $I_{i+1}$ are inputted into the Tracking Net (TN) to generate the motion and position map (MPM) $\tilde{C}_{i,i+1}$ using the

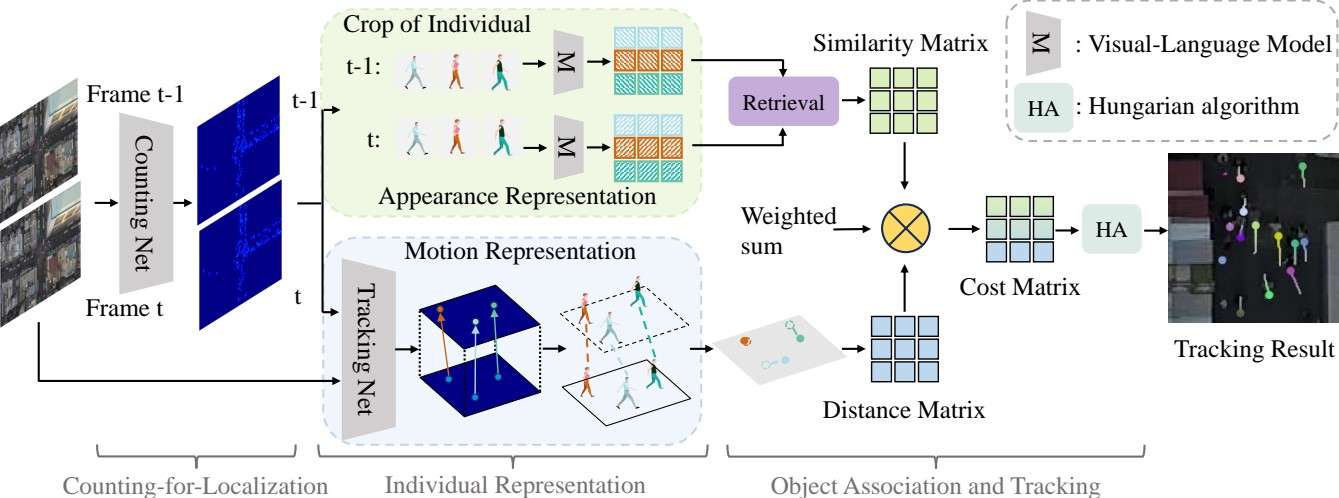

**Figure 2: DenseTrack is structured around three essential components: Localization, Individual Representation, and Association. Localization accurately determines the spatial positions of individuals in crowds through density maps. For Individual Representation, motion and appearance features are extracted by aligning density maps with motion and position maps (MPM) to provide motion cues, while the BLIP2 method is used to gather appearance cues. The Association component employs diffusion-based retrieval alongside a distance matrix derived from motion cues to facilitate precise inter-frame individual matching.**

following formula:

$$\tilde{C}_{i+1} = \text{TN}\left(I_i, I_{i+1}\right), \quad (9)$$

where MPM $\tilde{C}_{i+1}$ is a matrix with the shape of $(w, h)$, where $w$ represents the width of frame $I$ and $h$ represents the height of frame $I$. The values in the matrix represent a vector $\vec{o}$ describing the motion offset.

$$\tilde{C}_{i+1} = \begin{bmatrix} \vec{o}_{1,1} & \cdots & \vec{o}_{1,w} \\ \vdots & \ddots & \vdots \\ \vec{o}_{h,1} & \cdots & \vec{o}_{h,w} \end{bmatrix}. \quad (10)$$

To obtain the motion offset of the $j$-th individual in the $(i+1)$-th frame relative to the $i$-th frame, we retrieve the corresponding motion offset $\vec{o}_{x,y}$ at the respective position in $\tilde{C}_{i+1}$. Using the motion offset and coordinates, we can calculate the estimated position $\tilde{\text{cl}}_{i,j}$ of the $j$-th individual in the $(i+1)$-th frame as follows:

$$\tilde{\text{cl}}_{i,j} = \text{cl}_{i+1,j} + \vec{o}_{x,y}, \quad (11)$$

where $x$ and $y$ respectively represent the horizontal and vertical coordinates stored in $\text{cl}_{i,j}$, serving as indices to retrieve the motion offset stored in $\tilde{C}_{i,i+1}$.

## 3.4 Object Association and Tracking

This paper focuses on Multiple Object Tracking (MOT), which involves detecting multiple targets and assigning unique identities for trajectory tracking. After acquiring the positions of each individual in every frame, the task shifts to associating targets between consecutive frames. To enhance tracking accuracy, we integrate motion offsets and appearance features for inter-frame association.

Specifically, in the appearance feature association stage, we consider the appearances of the $k$-th individual in frame $i$ and the

$j$-th individual in frame $i+1$ to be inherently similar if they correspond. Drawing inspiration from the success of [45], we utilize the diffusion method (DM) to compare appearance representations $F_i$ across frames, akin to image retrieval. This process yields the similarity matrix $A^S_{i,i+1} \in \mathbb{R}^{p \times q}$, where $p$ represents the number of individuals detected in the previous frame $I_i$, and $q$ denotes the number of individuals detected in the subsequent frame $I_{i+1}$.

$$A^S_{i,i+1} = \text{DM}\left(F_i, F_{i+1}\right), \quad (12)$$

where the values in $A^S_{i,i+1}$ are represented as follows:

$$A^S_{i,i+1} = \{a^S_{k,j}\} = \begin{bmatrix} a^S_{1,1} & \cdots & a^S_{1,q} \\ \vdots & \ddots & \vdots \\ a^S_{p,1} & \cdots & a^S_{p,q} \end{bmatrix}, \quad (13)$$

where $a^S_{k,j}(k \in (1,p), j \in (1,q))$ represents the appearance similarity score between the $k$-th individual in frame $i$ and the $k$-th individual in frame $i+1$, ranging between 0 and 1.

Simultaneously, to ensure that the estimated positions $\tilde{P}_i$ of each individual appearing in frame $i+1$ closely align with their actual positions $\text{CL}_i$ in frame $i$, we construct a matrix $A^D_{i,i+1} \in \mathbb{R}^{p \times q}$. This matrix represents the estimated positions of each individual in frame $i+1$ relative to the actual positions of individuals in frame $i$:

$$A^D_{i,i+1} = \{a^D_{k,j}\} = \begin{bmatrix} a^D_{1,1} & \cdots & a^D_{1,q} \\ \vdots & \ddots & \vdots \\ a^D_{p,1} & \cdots & a^D_{p,q} \end{bmatrix}, \quad (14)$$

where $a^D_{k,j}(k \in (1,p), j \in (1,q))$ represents the Euclidean distance between the actual position $\text{CL}_{i,k}$ of the $k$-th individual in frame $i$ and the predicted position $\tilde{\text{CL}}_{i,j}$ of the $j$-th individual in frame

---

**Algorithm 1:** Inter-Frame Association for Tracking

---

**Input** : Localization of individuals in each frame,
$\mathrm{CL} = \{\mathrm{CL}_1, \mathrm{CL}_2, \cdots, \mathrm{CL}_N\}$; Cost matrix between two frames, $A^C_{i,i+1}$, for $1 \leq i < N$.

**Output** : Trajectories of individuals across frames, $T = \{T_1, T_2, \cdots, T_M\}$.

1   Initialize tracking for the first frame in the video:
2   **for** $j = 1$ **to** $\mathrm{Len}(\mathrm{CL}_1)$ **do**
3     Assign initial positions: $T_{j,1} = \mathrm{CL}_{1,j}$
4   **end**
5   **for** *each subsequent frame* $i = 1$ **to** $N$ **do**
6     Compute matching pairs using the cost matrix:
7     $\mathrm{ML}_{i-1,i} = \mathrm{HA}(A^C_{i-1,i})$
8     **for** *each match* $k = 0$ **to** $\mathrm{Len}(\mathrm{ML}_{i-1,i})$ **do**
9       Assuming the $id$-th trajectory is matched with the $u$-th individual in the $i$-th frame:
10      Update trajectories: $T_{\mathrm{id},i} = \mathrm{CL}_{i,u}$
11     **end**
12     If an individual $r$ in frame $i$ is unmatched, assign a new ID for a new trajectory:
13     $\mathrm{id}_c = \mathrm{Len}(\mathrm{ML}_{i-1,i}) + 1$,
14     $T_{\mathrm{id}_c,i} = \mathrm{CL}_{i,r}$
15   **end**

---

$i + 1$, as estimated in frame $i$:

$$a^D_{k,j} = \sqrt{(x - \tilde{x})^2 + (y - \tilde{y})^2}, \quad (15)$$

where $x$ and $y$ denote the coordinates of $\mathrm{CL}_{i,k}$ along the x- and y-axis, respectively, and similarly, $\tilde{x}$ and $\tilde{y}$ are defined.

From the preceding steps, both the similarity matrix $A^S_{i,i+1}$ and the distance matrix $A^D_{i,i+1}$ offer means to gauge the likelihood that individuals in two frames are the same. However, if solely relying on the similarity matrix, distance issues are overlooked, potentially resulting in the assignment of the same ID to spatially distant individuals. Conversely, if matching relies solely on the distance matrix, ID switches can happen within clusters of individuals due to highly similar distance cues, harming tracking outcomes.

Therefore, a synergistic approach of motion and appearance for inter-frame association is adopted, aiming to complement these two metrics by addressing different aspects of the assignment problem. Initially, the values in the distance matrix $A^D_{i,i+1}$ are rescaled to range between 0 and 1, resulting in the transformed distance matrix denoted as $\hat{A}^D_{i,i+1}$. To formulate the association problem, a weighted sum is employed to integrate both metrics, as follows:

$$A^C_{i,i+1} = (-\lambda)\,\hat{A}^D_{i,i+1} + (1 - \lambda)\,A^S_{i,i+1}. \quad (16)$$

Before combining the matrices, the distance matrix $A^D_{i,i+1}$ is multiplied by $-\lambda$ to adjust its influence. In the matching task, smaller values in $A^D_{i,i+1}$ suggest a higher likelihood of representing the same individual, while larger values in $A^S_{i,i+1}$ indicate a higher likelihood of representing different individuals.

After obtaining the cost matrix $A^C_{i,i+1}$, we employ the Hungarian algorithm (HA) to determine the optimal matches between frames using both metrics. This facilitates the establishment of associations across frames, enabling the deduction of each individual's trajectory in the video for every frame, denoted as $T$. The detailed procedure is outlined in Algorithm 1. Through the aforementioned operations, the trajectory $T$ is obtained, composed of the positions where each ID appears in every frame, completing the tracking process.

## 4 EXPERIMENTAL RESULTS

### 4.1 Dataset and Metrics

*4.1.1 Dataset.* Our experiments utilize DRONECROWD dataset [41], which includes 112 video clips from diverse scenes. These clips feature diverse lighting conditions (sunny, cloudy, or night), object sizes (diameters greater than 15 pixels or less or equal to 15 pixels), and densities (average object count per frame exceeding 150 or below 150). The dataset is captured using a high-definition camera at a resolution of $1920 \times 1080$, recording at 25 frames per second (FPS). The dataset provides annotations for the trajectories of 20,800 individuals and 4.8 million heads. Moreover, it segments the frame-by-frame images from the 112 video clips into 142 sequences, each containing 300 frames. These are further divided into 82 sequences for training, 30 for validation, and 30 for testing.

*4.1.2 Metrics.* In evaluating crowd tracking algorithms, we utilize temporal mean average precision (T-mAP) for trajectory accuracy, considering thresholds (T-AP@0.10, T-AP@0.15, T-AP@0.20) and a 25-pixel accuracy threshold to validate tracklets. Since our method outputs location points rather than precise bounding boxes, localization average precision (L-AP) is not applicable [41]. Instead, we employ mean absolute error (MAE) and root mean square error (RMSE) for localization performance, aligning with established crowd counting metrics. These metrics are chosen to comprehensively assess both the precision of tracking individual trajectories and the accuracy of localizing individuals within a crowd.

### 4.2 Implementation Details

We implement our method based on the PyTorch framework. To efficiently train FIDT [21], we choose the adaptive moment estimation (Adam) [15] optimizer. In training, the batch size is set to 16, and the crop size is set to 256. To streamline operations, we directly feed cropped head images sized at $20 \times 20$ into the unified interface offered by LAVIS[1] to acquire extracted image features. As we combine appearance and motion cues, we assign a weight of 0.9 to the parameter $\lambda$. This entire process is executed on the NVIDIA GeForce RTX 4090 platform.

### 4.3 Comparison with State-of-the-Arts

Tab. 1 presents a comparative analysis of tracking performance on DRONECROWD. STNNet [41] relies solely on motion-based methods, its performance is suboptimal as it may misidentify individuals in close proximity. In contrast, DenseTrack integrates both motion and appearance, mitigating this issue. It performs exceptionally well, achieving the highest T-mAP score of 39.44, excelling across all thresholds. Demonstrating outstanding tracking capability, especially in challenging environments, DenseTrack's strong performance at lower thresholds highlights effectiveness under less stringent conditions, while competitiveness at higher thresholds showcases reliability across diverse tracking scenarios.

### 4.4 Ablation Study

*4.4.1 Ablation Study on Density Localization.* Tab. 2 presents a comparison of direct counting-based human localization versus

---

[1] https://github.com/salesforce/LAVIS

**Table 1: Tracking performances on DRONECROWD; average T-mAP, and T-AP at each threshold (T-AP$_{0.10}$, T-AP$_{0.15}$, and T-AP$_{0.20}$). MOT and DCT stands for Multi Object Tracking and Drone-based Crowd Tracking, respectively. The best results are highlighted in bold.**

| Method | MOT | DCT | T-mAP | T-AP$_{0.10}$ | T-AP$_{0.15}$ | T-AP$_{0.20}$ |
|---|---|---|---|---|---|---|
| MCNN [48] | ○ | ● | 9.16 | 11.47 | 9.65 | 6.36 |
| CSRNet [19] | ○ | ● | 12.15 | 17.34 | 12.85 | 6.26 |
| CAN [22] | ○ | ● | 4.39 | 6.97 | 4.72 | 1.48 |
| DM-Count [38] | ○ | ● | 17.01 | 22.38 | 18.34 | 10.29 |
| STNNet [41] | ● | ● | 32.50 | 35.45 | 33.99 | 28.05 |
| Deep-OC-SROT [25] | ● | ○ | 28.39 | 30.84 | 28.52 | 25.81 |
| DenseTrack (Ours) | ● | ● | **39.44** | **47.48** | **39.88** | **30.95** |

**Table 2: Detection performances on DRONECROWD; The columns under "Counting" represent localization errors using only the counting network, while those under "Tracking" show errors refined through both the counting and tracking networks. The best results are highlighted in bold.**

| Method | Couting | | Tracking | |
|---|---|---|---|---|
| | MAE | RMSE | MAE | RMSE |
| STNNet [41] | **15.8** | **18.7** | 59.2 | 69.2 |
| MPM [13] | 22.1 | 31.5 | 22.1 | 31.5 |
| DenseTrack (Ours) | 20.3 | 21.4 | **19.2** | **29.0** |

**Table 3: Ablation studies investigate different factors influencing tracking performance, with each row depicting the impact of various solutions on tracking performance. "Counting" denotes tracking based solely on counting for localization and motion information tracking, "Appearance" represents appearance information, and "HA" stands for the Hungarian algorithm for matching. The best results are highlighted in bold.**

| Counting | Appearance | HA | T-mAP | T-AP$_{0.10}$ | T-AP$_{0.15}$ | T-AP$_{0.20}$ |
|---|---|---|---|---|---|---|
| ● | ○ | ○ | 2.90 | 3.45 | 2.95 | 2.29 |
| ● | ● | ○ | 37.46 | 45.59 | 37.80 | 28.99 |
| ● | ● | ● | **39.44** | **47.48** | **39.88** | **30.95** |

tracking-enhanced localization. When considering only counting, STNNet [41] outperforms other methods with MAE of 15.8 and RMSE of 18.7. However, upon integration of tracking, STNNet experiences a significant increase in errors, with MAE of 59.2 and RMSE of 69.2. In contrast, while DenseTrack initially shows slightly higher errors in counting alone compared to STNNet, with MAE of 20.3 and RMSE of 21.4, it substantially improves localization accuracy with tracking adjustments, achieving MAE of 19.2 and RMSE of 29.0. This highlights DenseTrack's effectiveness in leveraging tracking information to improve localization accuracy, thereby outperforming STNNet in the tracking-enhanced scenario.

*4.4.2 Ablation Study on Various Factors Performance.* To evaluate the contribution of each component to the enhancement of tracking performance, we present the results of tracking effectiveness after omitting certain steps, as detailed in Tab. 3.

Specifically, the first row employs only counting-based localization and motion tracking, resulting in relatively low T-mAP (2.90)

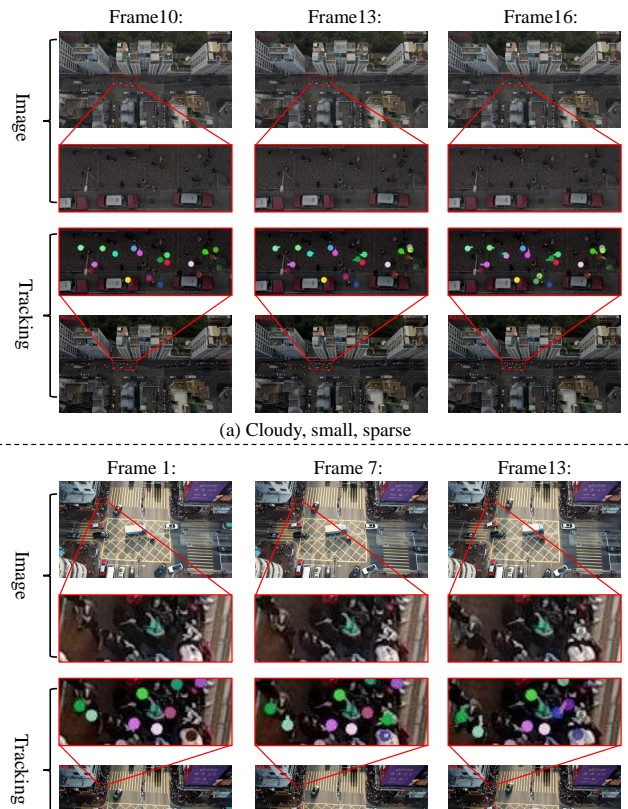

(a) Cloudy, small, sparse

(b) Sunny, small, Crowed

**Figure 3: Illustration of tracking under different conditions. (a) Sparse small objects in cloudy weather conditions. (b) Dense small objects in sunny weather conditions, with the same color representing the same individual.**

and T-AP at various thresholds (T-AP0.10: 3.45, T-AP0.15: 2.95, T-AP0.20: 2.29). Introducing appearance information in the second row leads to significant improvements across all metrics, particularly with T-mAP increasing to 37.46, and notable enhancements in T-AP thresholds (T-AP0.10: 45.59, T-AP0.15: 37.80, T-AP0.20: 28.99). However, the most significant performance boost is observed in the third row, where all factors are combined. Here, T-mAP reaches

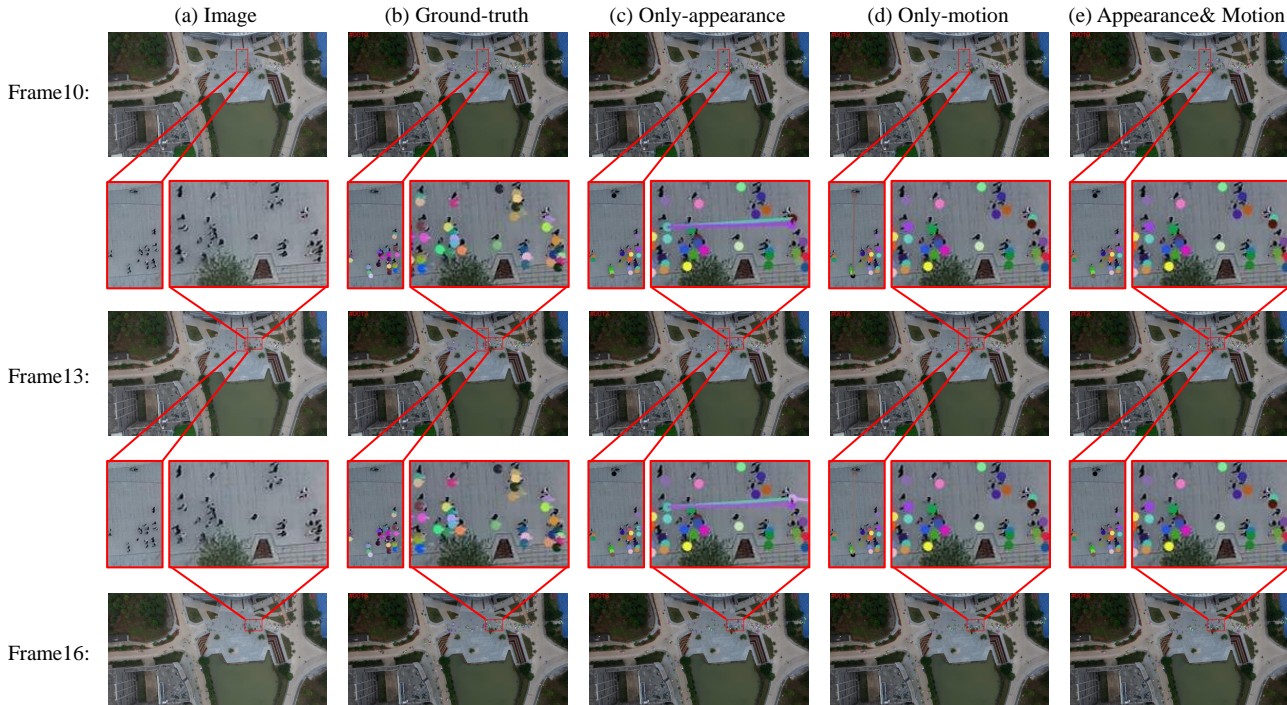

**Figure 4: Illustration of tracking performance using different strategies across frames 10, 13, and 16: (a) original aerial image, (b) ground-truth annotations, (c) tracking based solely on appearance, (d) tracking based solely on motion, and (e) tracking integrating appearance and motion. Insets magnify tracking results, showcasing the performance of each strategy.**

**Table 4: Ablation studies evaluating tracking performance using different VLMs: CLIP, BLIP, and BLIP2. The best results are highlighted in bold.**

| Method | T-mAP | $T\text{-}AP_{0.10}$ | $T\text{-}AP_{0.15}$ | $T\text{-}AP_{0.20}$ |
|---|---|---|---|---|
| CLIP [27] | 39.33 | 47.25 | 39.64 | 31.12 |
| BLIP [18] | 39.19 | 47.07 | 39.68 | 30.82 |
| BLIP2 [17] | **39.44** | **47.48** | **39.88** | **30.95** |

**Table 5: Ablation studies comparing the impact of distance measurement on the similarity matrix of appearances. Each row shows the performance using Cosine, Euclidean, and Diffusion distance. The best results are highlighted in bold.**

| Retrieval Method | T-mAP | $T\text{-}AP_{0.10}$ | $T\text{-}AP_{0.15}$ | $T\text{-}AP_{0.20}$ |
|---|---|---|---|---|
| Cosine | 39.32 | 47.02 | 39.88 | **31.05** |
| Euclidean | 39.33 | 47.13 | **39.91** | 30.95 |
| Diffusion [45] | **39.44** | **47.48** | 39.88 | 30.95 |

39.44, and T-AP thresholds peak (T-AP0.10: 47.48, T-AP0.15: 39.88, T-AP0.20: 30.95). These results underscore the critical role of considering appearance information and employing a matching algorithm for achieving optimal tracking performance in DroneCrowd.

*4.4.3 Ablation Study on Visual Representation.* Tab. 4 presents the tracking performance of different visual-language models (VLMs), showcasing their effectiveness in improving tracking accuracy. While all methods exhibit notable performance, BLIP2 [17] stands out as the top performer, achieving a T-mAP score of 39.44. This result underscores the efficacy of BLIP2 in enhancing tracking performance compared to other VLMs like CLIP [27] and BLIP [18]. The consistent superiority of BLIP2 across various precision thresholds highlights its robustness and effectiveness in capturing intricate visual and language cues for more accurate tracking. This analysis suggests that BLIP2's architecture incorporates beneficial features that contribute to its superior performance, making it a promising choice for tracking tasks in diverse scenarios.

*4.4.4 Ablation Study on Retrieval Method.* Tab. 5 provides a comprehensive comparison of the impact of different retrieval methods on tracking performance. Across all evaluated distance metrics, Cosine, Euclidean, and Diffusion, we observe notable improvements in tracking accuracy. While each method demonstrates effectiveness, the diffusion retrieval method stands out with the highest T-mAP score of 39.44 and $T\text{-}AP_{0.10}$ score of 47.48. This signifies its superior performance in associating individuals across frames. The results indicate that leveraging appearance-based retrieval methods, especially through diffusion, notably improves tracking accuracy.

## 4.5 Qualitative Analysis

*4.5.1 Analysis of Tracking Performance in Varied Conditions.* Fig. 3 showcases the capability of DenseTrack to effectively manage complex tracking scenarios. DenseTrack reliably identifies and tracks

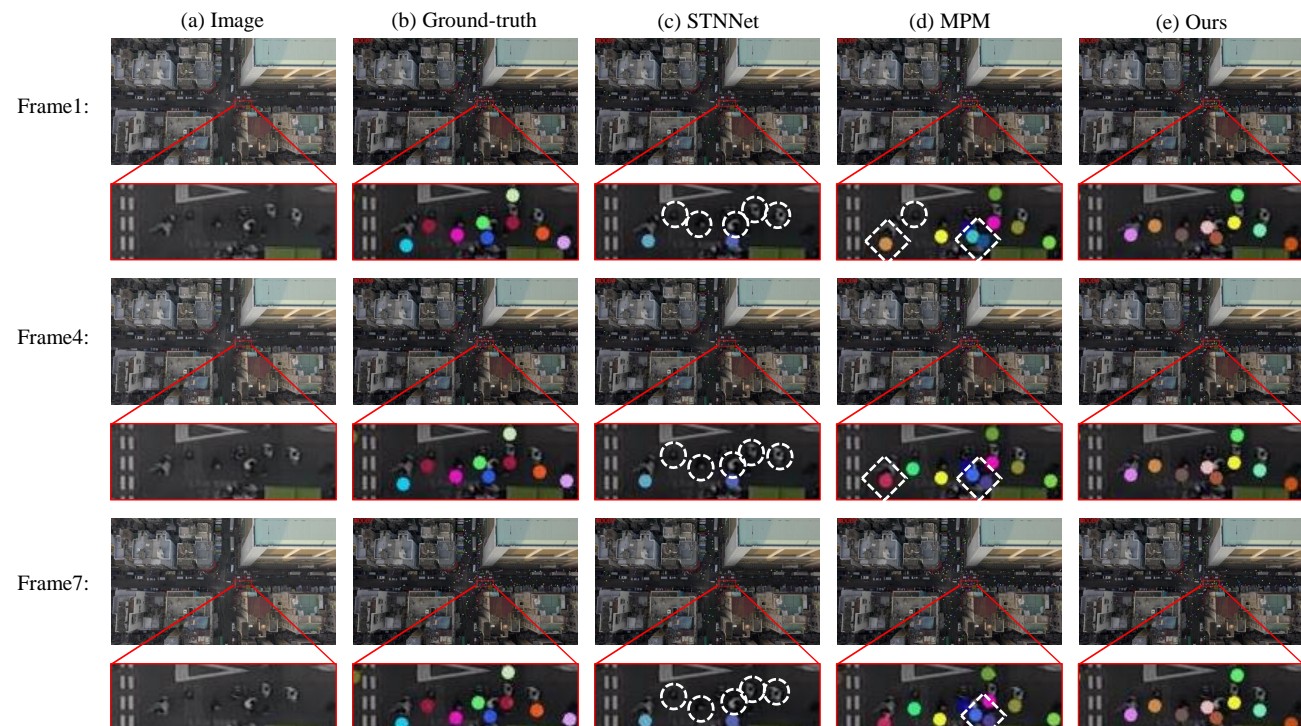

**Figure 5: Comparison of different tracking methods across frames 1, 4, and 7: (a) original surveillance footage, (b) ground-truth annotations, (c) tracking results from STNNet, (d) tracking results from MPM, and (e) our DenseTrack results. False negatives are marked with white dotted circles, and tracking switch errors with white rectangles. Insets provide a detailed view of tracking discrepancies, using consistent color coding to identify each individual.**

individuals across various environmental challenges, maintaining its accuracy even under conditions of cloud cover and high crowd density. This performance demonstrates the robustness of integrating motion and appearance cues within DenseTrack, allowing for precise tracking that is largely unaffected by scene complexities. The framework's adeptness in such diverse conditions underscores its advanced design and suitability for varied aerial applications.

*4.5.2 Analysis of Different Tracking Strategies.* Fig. 4 visually compares different tracking strategies to highlight the effectiveness of integrating appearance and motion information. The appearance-only strategy (Fig. 4(c)) though it accurately identifies all individuals and tracks most correctly, suffers from errors over long distances. These are significantly reduced when motion information is included. The motion-only strategy (Fig. 4(d)) avoids long-distance errors but tends to misidentify nearby targets. By combining both approaches, the integrated method (Fig. 4(e)) effectively balances distance considerations, minimizes errors with proximal targets, and thereby achieves optimal tracking performance.

*4.5.3 Analysis of Tracking Performance.* Fig. 5 offers an insightful comparative analysis, shedding light on the efficacy of our Dense-Track algorithm when juxtaposed with two prominent counterparts: STNNet [41] and MPM [13]. Each snapshot within the figure unveils distinct facets of the localization challenges and tracking discrepancies inherent in these methodologies. Examining STNNet's

depiction (Fig. 5(c)), significant localization errors are evident, highlighting the pivotal role of robust localization techniques in tracking precision. Conversely, the MPM-based approach (Fig. 5(d)) shows some improvement but remains prone to occasional false detections. In contrast, the DenseTrack method (Fig. 5(e)) notably improves localization accuracy and tracking precision. Its ability to accurately identify and track individuals across various scenarios underscores its effectiveness in addressing complex tracking challenges.

## 5 CONCLUSION AND DISCUSSION

In this work, we present DenseTrack, a novel tracking-by-counting method that enhances drone-based crowd monitoring by integrating appearance and motion cues. We construct a cost matrix combining a density-aware appearance similarity matrix with a cross-frame motion distance matrix, and apply the Hungarian algorithm to achieve robust tracking outcomes. DenseTrack demonstrates competitive performance in crowded drone surveillance environments. To the best of our knowledge, this is the first implementation to synergistically use both appearance and motion information for drone-based crowd tracking.

**Limitations and Future Work.** Despite its strengths, Dense-Track is not fully optimized for all environmental conditions and tends to underperform in low-light or cloudy scenarios. Future research will focus on enhancing its adaptability and effectiveness in a wider array of challenging surveillance contexts.

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
