# OpenReview forum: "DenseTrack: Drone-based Crowd Tracking via Density-aware Motion-appearance Synergy"
_acmmm.org/ACMMM/2024/Conference — MM2024 Poster_

### Official Review · Reviewer_rAE2 · 2024-05-24

**Rating:** 3
**Confidence:** 3

**Summary:**

This paper adopts crowd counting localization to achieve accurate localization of small targets, while using visual language models to obtain the appearance features of targets to overcome the disadvantage of losing appearance features in crowd counting localization. Finally, a good tracking effect was achieved by integrating motion features and appearance features.

**Strengths:**

1. Combining count-based detection and density-based individual recognition to achieve accurate localization while maintaining feature extraction of objects.

2. Integrating visual language models to enhance the process of individual recognition in density maps significantly improves the descriptive ability of density maps, resulting in a more detailed and accurate presentation of individuals in crowded scenes.

3. Synergistically combining motion and appearance cues to achieve tracking, complementing each other's strengths and improving tracking performance.

**Limitations:**

1.The identification made in Figure (d) should be indicated and clarified in the introduction.

2.Is $C\tilde{L}_{\mathrm{i-1}}$ obtaining the prediction result for the next frame based on the coordinates of the previous frame near line 255? Is there an error in the subscript here?

3.After matching in Hungary, how are the detections and trajectories that are not matched handled? There is no explanation provided in the text.

4.How to select the values of hyperparameters in Equation 16?

5.Qualitative experimental results can be compared under different thresholds.

6.In qualitative analysis, it is necessary to indicate the target of misidentification in Figure 4 (d).

7.Some reference formats are incorrect, it is recommended to carefully check (e.g. 38).

**Suitability:**

2

---

### Official Review · Reviewer_QCVr · 2024-05-24

**Rating:** 3
**Confidence:** 4

**Summary:**

This paper introduces a novel framework to address the challenges of tracking individuals in drone-based crowd monitoring scenarios. The framework integrates motion and appearance cues to enhance tracking accuracy, particularly for small and densely packed objects. It employs a crowd counting network for localization, utilizes a visual-language model for appearance representation, and applies the Hungarian algorithm for accurate object association across frames. Experiments conducted on the DroneCrowd dataset demonstrate DenseTrack's superior performance over existing methods.

**Strengths:**

(1)This paper presents an innovative approach to incorporate motion and appearance cues in a crowd counting localization paradigm.

(2)The proposed method shows its effectiveness by achieving state-of-the-art results on the DroneCrowd dataset.

(3)The proposed method shows a robust performance under a variety of conditions which include varying light and density.

**Limitations:**

(1) The experimental results is unconvincing for the method was validated on only one dataset. It would be better to conduct a broader comparison using more datasets such as UCSD, LHI, Fish, Cell, PETS2009, or CroHD.

(2)There are unclear explanations about adopting T-mAP as the evaluation metric instead of the standard MOT tracking metric, and many related works are missing in the benchmarking, that are mentioned in Related Work such as [28] and [35].

(3)The authors simply used the visual language model BLIP2 in their approach without incorporating textual information, which may not truly reflect the claimed integration of visual language models.

**Suitability:**

2

---

### Official Review · Reviewer_j8rF · 2024-05-26

**Rating:** 3
**Confidence:** 3

**Summary:**

In this paper, they introduce the Density-aware Tracking framework, which advances the counting-based localization by incorporating both motion and appearance cues. They extract detailed appearance information from density maps for precise individual identification and correcting motion discrepancies. Furthermore, they utilize visual-language models to derive intricate appearance features from density maps, ensuring accurate characterizations. The experimental result on DRONECROW dataset demonstrate the effectiveness and performance of the approach.

**Strengths:**

1) The writing of this paper is good. The whole structure is well-organized and logical, makes the content accessible to readers.
2) The design of experiments is thoughtful.

**Limitations:**

1) The theoretical innovation of this article is insufficient and appears to be more of an integration of existing methods. Including motion and position map, visual-language model, tracking network, and Hungarian matching, are not original contributions of this article. Furthermore, simultaneously use both appearance and motion information for multi-object tracking is a common method. I don’t think it can be considered an innovative point, even in the field of drones.
2) There are some symbol errors, for example, in line 449, the correct content should be “score between the k-th individual in frame i and the j-th individual in frame i+1”. Please check it carefully.
3) For each metric in the table, it will be better use symbol ↑/↓ to represent that the higher / lower value denotes better performance

**Suitability:**

2

---

### Meta-Review · Area_Chair_P3Vq · 2024-06-26

**Recommendation:** Accept (Poster)
**Confidence:** 5

**Metareview:**

The paper introduces the density-aware tracking framework, which enhances counting-based localization by integrating motion and appearance cues. It employs visual-language models to derive detailed appearance features from density maps, ensuring accurate individual identification and motion correction. The approach demonstrates effectiveness on the DRONECROW dataset, showing improved performance and robustness.

While the integration of existing methods such as motion and appearance cues and visual-language models is not entirely novel, the combination is well-executed and provides significant improvements in tracking accuracy. The framework's performance is validated through thoughtful experimental design, though broader comparisons with additional datasets would strengthen the results. The authors addressed key concerns in their rebuttal.